# Hydrogen Bonding of Trialkyl-Substituted Urea in Organic Environment

**DOI:** 10.3390/molecules30071410

**Published:** 2025-03-21

**Authors:** Zuzana Morávková, Jiří Podešva, Valeriia Shabikova, Sabina Abbrent, Miroslava Dušková-Smrčková

**Affiliations:** Institute of Macromolecular Chemistry, Czech Academy of Sciences, 162 00 Prague, Czech Republicm.duskova@imc.cas.cz (M.D.-S.)

**Keywords:** urea, IR, H-bonding, (poly)urea, aspartate

## Abstract

Urea groups appear in many biomolecules and polymers. They have a significant impact on the properties of the materials because of their inherent strength and for their ability to participate in hydrogen bonds. Typically, in classical urea-based polymer materials, the urea groups occur in their *N*,*N*′-disubstituted state. Recently, bis-aspartates have been introduced as a novel type of hindered amine resins providing, upon crosslinking with (poly)isocyanates, the polyurea–polyaspartate thermosets (PU-ASPE) for coatings, sealants, polyelectrolytes, and other applications. These materials contain *N*,*N*′*N*′-trisubstituted urea linkages in their structures. However, the infrared (IR) characterization of trisubstituted urea groups has not been documented in sufficient detail. Consequently, studies on the structure of aspartate-based polyurea materials often rely on data from *N*,*N*′-disubstituted ureas, which can lead to inaccurate conclusions. This study presents a detailed evaluation of the possible urea H-bonding states, focusing on the difference between the di- and trisubstituted species. Particularly, the attributions of the IR spectra to urea-based hydrogen bonding states are presented both in neat materials and their solutions. To systematize this study, we initially focus on a simple trisubstituted urea model system, tributyl urea (3BUA), and compare its spectral response with disubstituted *N*-butyl-*N*′-cyclohexyl urea (1B1CHUA) and trisubstituted *N*-butyl-*N*′,*N*′-dicyclohexyl urea (1B2CHUA), to elucidate their hydrogen-bonding fingerprints. This research provides a thorough understanding of the IR response of the di- and trisubstituted urea species and their structural characteristics in urea-containing materials.

## 1. Introduction

The urea group is an important building block in a variety of chemical species, ranging from natural biomolecules, such as vitamins or enzymes, to synthetic pharmaceuticals and high performance polymers. The biological activity of a molecule is closely related to its conformation in space (a tertiary structure in case of biomacromolecules), which is dictated by non-covalent interactions, including H-bonds. The urea segment, similarly to an amide, can form one of the strongest H-bonds available. Typically, in aqueous systems, the disubstituted ureas exert a strong tendency to self-assembly and function also as active gelators [1]. Urea groups are also known for bringing high cohesion and remarkable chemical stability into polymer materials in organic environment [2].

In materials engineering, several important thermosets possess urea groups in their chemical structure. For example, urea–formaldehyde resins (UF), a well-established type of amino resins, are known for their high tensile strength, outstanding heat resistance, and low moisture absorption, all at the same time [3]. These properties make them ideal for long-term applications, such as adhesives, plywood, and particleboards. In the linear polymeric backbone of UF resins, the urea groups are disubstituted and alternate with methylene groups.

In conventional polyurethane (PURs) systems, diamines can be used as chain extenders. This modification results in the formation of linear polyurea regions, which typically act as hard segments reinforcing the material [4].

During the formation of polyurethanes, namely polyurethane coatings, a certain portion of the NCO groups in the initial isocyanate component may react with residual water (typically ranging from one to ten per cent). This reaction eventually yields disubstituted urea groups joining two isocyanate units and thus forming isocyanate–urea segments [5] (polyurea, PUA), modifying the polyurethane matrix. These PUA segments presumably affect the local properties of the matrices on a microscopic level, such as their strength and value of glass transition temperature, *T*_g_.

In water-blown polyurethane foams (PUFs), part of the NCO groups is deliberately used for additional reactions with water, so, analogously to the moisture-cure of isocyanates mentioned above, a primary amine group is formed that subsequently reacts with another NCO, leading to the disubstituted urea groups [3]. In PUFs, the urea groups contribute to material rigidity and hardness and their concentration is the main classification parameter for the division into soft and hard PURs [6].

Polyurea-based polymers (PUAs) are prepared by the reaction between isocyanate and amine groups of given precursors [3]. They are applied in coatings, sealants, and elastomers, particularly in applications where a fast-curing process is crucial. For example, cycloaliphatic diamines are often employed as co-reactants with (poly)isocyanates in polyurea coatings [4]. To reduce a too-fast cure rate between primary amine and isocyanate, the functional groups of either component can be blocked, allowing for a controlled curing rate. In coatings, blocking of the amine group is typically achieved by its addition to the carbonyl group of ketone or aldehyde leading to ketimine or aldimine [7]. These compounds decompose during the reaction with isocyanate and the blocking agent evaporates during the film forming process.

Certain types of trisubstituted ureas bearing aromatic and bulky substituents revealed the ability to undergo acyl substitution, decomposing into isocyanate and amine moieties in situ. These types of urea were reported as masked isocyanates [8]. The ability of the hindered urea groups to transition into isocyanates and amines has been further utilized in research on covalent dynamism. The transformation mechanism is mediated by hindered urea groups and promises the development of PUA and poly(urethane-urea) self-healing materials [9].

A novel class of amine resins, known as bis-aspartates or aspartates (ASPE, Figure 1), with amine group hindered with an ester group, has recently emerged. The bis-aspartates possess secondary amine groups located near the ester residues. The carbonyl groups of esters exert an electron-withdrawing effect, reducing the basicity of the amine’s hydrogen atom. This results in a significantly slower rate of cure with isocyanates, as compared to that of primary and simple secondary amines. Despite this reduced reactivity, bis-aspartate amines remain more reactive than the hydroxyl groups present in polyols used for the preparation of polyurethanes. Thus, heavy metal-based catalysts used to cure the resin can be reduced or even avoided, enabling an environmentally friendlier process. Furthermore, bis-aspartates exhibit lower viscosity, thus permitting smaller solvent content in curing mixtures. Beyond the environmentally beneficial aspects of their faster and cleaner cure, these resins impart enhanced chemical and weathering resistance, improved mechanical strength, superior adhesion, and, notably, superior corrosion protection to the final material, making them particularly suitable for high-performance coatings [10].

H-bonds influence numerous aspects of any system, from their effect on physical and mechanical properties to their interference with reaction kinetics when H-bonded moieties are involved. A typical example of this specific effect involving urea groups is the hydrogen bond between the amine group and the ester carbonyl, which causes a decrease in reactivity of the aspartate [11]. We have recently revisited a unique phenomenon of PU-ASPE curing: the formed trisubstituted urea groups in the ASPE system readily undergoing a subsequent reaction with an adjacent ester group while forming cyclic or linear ureido–carbonyl structures NH–CO–NH–CO and releasing one molecule of alcohol [7,10]. The detailed mechanism of this reaction has been under investigation, however, the influence of H-bonding exerted by urea groups and adjacent ester on the propensity of this transition can be presumed. The third example is the self-healing of trisubstituted urea: the C–N′ bond in *N*,*N*,*N*′-trisubstituted urea is destabilized when one of the N-substituents is bulky, which leads to the dissociation into a secondary amine and isocyanate [9,12]. The proposed mechanism is based on the hydrogen transfer from the NH group to another nitrogen; therefore, the participation of the NH group in hydrogen bonding can influence the reaction equilibrium [9,12,13].

### Characterization of Urea Species by IR

Urea species are commonly studied using IR. Due to the intensity of the C=O stretching vibration and its sensitivity to its surroundings, the carbonyl stretching region is typically the only feature analyzed by most authors [14,15]. The urea carbonyl stretching vibration is very sensitive to substitution on the neighboring nitrogen atoms [15], hydrogen bonding [16], as well as to solvent effects [15,17]. Most combinations of substituents on urea nitrogens are thoroughly described in the literature, including simple urea, monosubstituted, *N*,*N*- and *N*,*N*′-disubstituted, and tetrasubstituted forms [15]. However, the spectroscopic studies of trisubstituted urea have, so far, not considered the hydrogen bonding state of the urea groups [7,9,12,18], although the trisubstituted urea linkage is a key feature in novel polyurea materials [10,19].

The most studied form of urea is the *N*,*N*′-disubstituted species. As already mentioned, this group forms the linkage in classical PUA. Three distinct bands can be observed in the carbonyl region in the spectra of solid PUA: “free” urea band around 1690 cm^−1^, “disordered” urea band around 1660 cm^−1^, and “ordered” urea band around 1635 cm^−1^ [16,20]. The attribute “free” refers to the state of urea where the carbonyl group is free of hydrogen bonds while the H-bonding state of the N–H groups is not specified [16]. It was identified in diluted urea solutions or PUA melts [21]. The terms “disordered” and “ordered” are derived from the observations on solid PUA with soft and hard segments. The “disordered” IR band prevails in the spectra of soft polymer domains and in liquid ureas, while the “ordered” urea band appears in hard polymer domains. This led to the denomination of the “disordered” band to the short-range order and the “ordered” band to the long-range order [16,22].

In the IR studies of the urea linkages in PU-ASPE, the presence and state of trisubstituted urea groups have so far been either supported by the knowledge of *N*,*N*′-disubstituted urea [23,24,25] or not discussed in any detail. Therefore, the urea carbonyl stretching bands for trisubstituted urea have not yet been attributed in the literature.

The exact evaluation of the total content of urea, as well as its H-bonded state, is crucial for understanding the reaction mechanism in aspartate-based materials—both the chemical processes ongoing during curing and the influence of H-bonding on the mechanical properties of the final materials. Therefore, an effort will be exerted to correctly identify the corresponding bands and clarify the assignment of trisubstituted urea groups for future use. To achieve this, we will first analyze a model system—pure tributyl urea (3BUA) and its solutions—and evaluate the effect of H-bonds on its carbonyl and N–H stretching bands. As a next step, two commercially available molecules with di- and trisubstituted urea groups will be analyzed.

## 2. Results and Discussion

### 2.1. Background on the Urea IR Spectroscopy

The most studied vibration of the urea group is its carbonyl stretching (“amide I” appearing in the region 1615–1705 cm^−1^) [26]. N–H stretching (“amide A” above 3000 cm^−1^) [26] and N–H deformation coupled with C–N stretching (“amide II”, 1515–1605 cm^−1^) [26] bands are also relatively well resolved. However, they are rather broad and their positions are not responsive enough to the chemical environment of the urea group to enable a detailed analysis. In fact, only the free and H-bonded N–H have been distinguished in the N–H stretching region [14,17,27]. Even though detailed analysis of the N–H stretching region is rather complex [27,28], it holds information complementary to the carbonyl region [27,29]. The N–C–N stretching vibrations (antisymmetric in the region 1360–1300 cm^−1^ and symmetric in the region 1190–1140 cm^−1^) [26] can overlap with many other bands of common vibrations, such as C–O and C–C (skeletal) stretching [26] and are not reliably resolved. The relevant urea bands are summarized in Table 1.

As mentioned above, the position of the urea carbonyl stretching vibration depends on the substitution on the nitrogen atoms. The position of the carbonyl stretching band shifts to lower wavenumbers in the order unsubstituted → monosubstituted → *N*,*N*′-disubstituted → *N*,*N*-disubstituted → *N*,*N*,*N*′,*N*′-tetrasubstituted, because the –NH_2_ group has a higher electron donating capability compared to –NHR or –NR_2_ [15]. Harnagea et al. used inverse ionization potentials of the –NH_2_, –NHR, and –NR_2_ groups (9.0, 8.2, and 7.9 eV, respectively) as a measure of their H-donating capability [15]. If we adopt such calculations, the carbonyl stretching vibration of trisubstituted urea should lie around 1650 cm^−1^ (in ethanol), which corresponds to halfway between the *N*,*N*′-disubstituted and tetrasubstituted urea carbonyl stretching positions. In this particular work by Harnagea et al., [15] small model molecules in a variety of solvents were studied. Upon dilution in different solvents, the position of the C=O stretching band can change by ~70 cm^−1^ [15], thereby solutions in the same solvent shall generally be used for comparative studies. However, in PUA, typically, solid state spectra are discussed.

As pointed out above, three distinct carbonyl stretching bands for the disubstituted urea species are identified: “free” urea carbonyl band around 1690 cm^−1^, “disordered” urea carbonyl band around 1660 cm^−1^, and “ordered” urea carbonyl around 1635 cm^−1^ (Table 1) [16,20]. Some authors have observed a shoulder around 1700 cm^−1^ [27], its origin being explained in several different ways. Coleman et al. [27] proposed it belongs to urea monomer or oligomers formed at higher temperatures. Marcos-Fernandez et al. discussed the effect of the H-bond incorporating the N–H group on the carbonyl band position where the shoulder ~1700 cm^−1^ was assigned to a completely free urea, while the main “free” band around 1690 cm^−1^ belongs to urea with free carbonyl, but H-bonded through N–H group(s) [14].

The first atomistic explanation of the origins of the “ordered” and “disordered” urea bands appeared after the discovery of bifurcated H-bonds, where two N–H hydrogens of one urea group are bonded to the carbonyl group in another molecule. The term “ordered urea” was assigned to this H-bonded state (Figure 2) [14,27,31]. The “disordered” urea band was then assigned to urea molecules where the hydrogen bond is realized only through one N–H. However, “ordered” and “disordered” carbonyl stretching bands were also identified in urethanes, where the bifurcated structure is not possible [16]. Clearly, the bifurcated H-bond is not the defining parameter of the “ordered” H-bond state in the amide-class of molecules. It should be noted that the geometry of the H-bonded urea in the “ordered” and “disordered” state should be the same, because the envelope of their N–H stretching regions is identical [27]. Instead, Coleman et al. explained the difference with dipole–dipole interaction among the aligned carbonyl groups in the “ordered” state that lowers the carbonyl stretching vibration wavenumber [27]. The cooperativity of H-bonding and the difference between the isolated H-bond dimer and ordered array is commonly accepted [2]. According to Marcos-Fernandez et al., in *N*,*N*′-disubstituted urea, only the bifurcated H-bonds occur, but their band positions will differ depending on the location of the carbonyl bond in the stack of the ordered ureas (Figure 2). Inside the stack, the carbonyls are H-bonded through N–H…O=C sequence with carbonyl absorption around 1635 cm^−1^, while carbonyls at the top of the stack (urea groups with free carbonyl and H-bonded N–H groups) represent the “free” urea at around 1690 cm^−1^, and carbonyl bonded at the bottom of the stack (urea molecules with N–H groups not attached to another urea and H-bonded through the carbonyl) represent the “disordered” urea absorption around 1660 cm^−1^ [14].

In aspartate-based systems, two urea carbonyl bands are observed around 1625 and 1640 cm^−1^ [10,24]. In the literature, they have been assigned to the “ordered” and “disordered” H-bonded urea [23,24,25]. The region of “free” urea is often masked by the stretching band of the isocyanurate ring of the cyclotrimerized isocyanate crosslinker [10]. Therefore, no conclusion could be made about the presence of urea free from hydrogen bonds in an aspartate—isocyanurate crosslinking system; however, only the two bands were taken into account by the authors without deeper discussion of the actual state of the urea groups [23,24,25].

When studying a model aspartate system with monofunctional isocyanate (Figure 1), where the isocyanurate ring is not present, no absorption was observed around 1690 cm^−1^. This would imply that no free carbonyl groups were present in the system (all should be hydrogen bonded, either ordered or disordered). However, when assessing the balance of the H-bond donors and acceptors in this system, the donors were only represented by the amine groups of both the aspartate and the urea (Figure 1), where, during the addition reaction with isocyanate, the total amount of NH groups remained constant. The available H-bond acceptors are (1) the urea carbonyls, (2) aspartate ester carbonyls and ether oxygens (two ester groups per one aspartate-amine), and (3) possibly solvent molecules (butyl acetate in the mentioned case). Clearly, the total count of H-bond acceptors is in excess, which would imply the presence of free urea carbonyls. This contradiction suggests that the free urea response could appear elsewhere in the spectrum.

The urea carbonyl is a stronger H-bond acceptor than the ester carbonyl, so the H-bond among urea groups should be preferred [32,33]. On the other hand, the steric effect may favor other H-bond acceptors. For example, in the PU-ASPE network, (1) the mobility of the polymer segments may be hindered by crosslinking and thus the formation of H-bonds between two urea groups is impaired and (2) urea groups are sterically hindered by the maleate residue, making it less likely for them to reach another molecule for H-bonding.

To sum up, in the trisubstituted ureas, we expect to observe (1) the urea carbonyl free from H-bonds and (2) the “disordered” H-bonded structure. The “ordered” urea band, on the other hand, is not expected in trisubstituted ureas, and in PU-ASPE networks specifically. This correlates with the observation of just two urea-carbonyl stretching bands in the spectra of PU-ASPE species—1625 (H-bonded) and 1640 cm^−1^(free) [10,23,24,25].

### 2.2. Tributyl Urea as a Model Molecule

To prove the proposed assignment of the two observed PU-ASPE carbonyl bands by experiment, a model molecule, tributyl urea (3BUA, Figure 3), was synthesized. The product (2.23 g, yield 96%) contained, according to the GC/MS method, 99.06% of 3BUA (Appendix A). The molecular structure was confirmed by NMR (Appendix A). It was studied in three states—pure, diluted in a solvent acting as hydrogen-bond acceptor (ethyl acetate, EA), and in a solvent acting as hydrogen-bond donor/acceptor (isopropanol, IP) (Appendix A). Inevitably, individual urea carbonyl stretching bands (free 1635–1645 cm^−1^, disordered 1615–1630 cm^−1^, see Table 1) shift due to solvent effect, however, these effects are not the focus of this study.

In the neat 3BUA, the carbonyl stretching region consists of a band at 1621 cm^−1^ with only a faint shoulder at 1660 cm^−1^ (Figure 4a, Table 1). Upon increasing dilution with both selected solvents, the band at 1621 cm^−1^ decreases in intensity (Figure 5b,d) and shifts towards 1629 cm^−1^ (Figure 4b,c and Figure 5a,c). Upon dilution with EA, a shoulder around 1665 cm^−1^ increases and dominates the region in highly diluted solutions (Figure 4b and Figure 5b). EA, being a hydrogen bond acceptor, competes with the urea carbonyl for the urea NH hydrogens. Thereby, the typical urea H-bond N–H⋯O=C should be at least partially replaced by a H-bond with the solvent ester group (carbonyl or etheric oxygen): N–H⋯O(ester). On the contrary, upon dilution in IP, the main 3BUA band (~1626 cm^−1^) still dominates the region while the shoulders at 1663 and 1600 cm^−1^ increase only slightly (Figure 4c and Figure 5d). As IP can serve both as a hydrogen bond donor and acceptor, it competes with the urea carbonyl for available hydrogens, thus possibly leaving some of the urea carbonyls free from H-bonds, while at the same time, it offers additional hydrogens to the urea carbonyls. Because IP is a smaller molecule compared to 3BUA, and its OH group is well accessible, it is probably sterically easier for the urea carbonyl to bind to the IP OH-group than to the NH group of another urea unit. The OH group can even have some spatial freedom within the H-bond, which allows for dipole-alignment between the carbonyl and the OH, thus explaining the observed shift of the carbonyl stretching band to 1600 cm^−1^. For the whole spectra of the carbonyl region in 3BUA solutions, see Appendix A.

Such change in H-bonding should have an effect also on the N–H vibrations (Figure 6 and Appendix A, Table 1). Indeed, the N–H stretching vibration of neat urea, observed at 3345 cm^−1^, is seen to shift towards 3370 cm^−1^ upon dilution with EA, while an additional band appears around 3445 cm^−1^, eventually replacing the original one (Figure 6a,b). The NH group is a strong H-bond donor and should be H-bonded if possible [34], thereby the band of free NH is not expected. The urea carbonyl is a stronger H-bond acceptor than ester, thus, the formed H-bond should be stronger. Therefore, the new band at 3445 cm^−1^ may be attributed to the weaker H-bond with the ester group. This region could not be analyzed for solutions in IP due to the overlap with the OH stretching band of alcohol.

The amide II band, on the other hand, is well observed in both solvents and is seen to shift in opposite directions (Figure 5a,c, Appendix A, Table 1). With increasing dilution in EA, it shifts towards 1523 cm^−1^ (1527 cm^−1^ in dilution 1:3), while in IP, it shifts towards 1540 cm^−1^ (1534 cm^−1^ in dilution 1:3). The amide II band (NH deformation coupled with CN stretching) should shift to lower wavenumbers on H-bonding [26]. This means that in IP, the strength of the H-bonding involving urea NH groups is higher than in neat 3BUA and even more than in EA. This can relate to multiple H-bonds directed to a single NH group (multiple alcohol OH groups).

The effect of the H-bonding on the vibrational responses of both pure solvents is small and not straightforward to interpret. The carbonyl stretching of the EA ester group has two components (1729 and 1737 cm^−1^) (Figure 4b and Figure 5a). These components cannot be assigned to any H-bond state as there should not be any strong H-bonds in pure solvent. With the addition of urea, the ratio of these components changes, but there is no clear trend. This is not too surprising, as ester carbonyls are known not to be very sensitive to H-bonding [26]. The IP-dilution series gives even less information on the state of IP itself. The OH stretching band remains unchanged, the OH deformation band overlaps with CH deformations, and only in the alcohol C–O stretching region do we observe a slight shift in one of the bands from 1127 to 1129 cm^−1^ (Appendix A). We can use these observations only as an indication of a change but cannot deduce any details about the nature of the process.

To sum up, the dilution dependencies suggest that the main band of 3BUA at 1621 cm^−1^ belongs to the C=O group of urea, H-bonded to the NH group of another urea, C=O ⋯H–N), while the band ~1660 cm^−1^ can be assigned to the free urea groups (free from H-bond on the urea carbonyl, but possibly with the NH group H-bonded as N–H⋯X), and finally, the band at 1600 cm^−1^ can be related to the urea carbonyl H-bonded to alcohol OH (Table 1).

### 2.3. Comparison of Model Molecules

Two commercially supplied urea species, one disubstituted, *N*-butyl-*N*′-cyclohexyl urea (1B1CHUA, Figure 3), and one trisubstituted, *N*-butyl-*N*′,*N*′-dicyclohexyl urea (1B2CHUA, Figure 3), were also investigated by IR to assess the effect of the substituent sizes (Figure 2, Figure 7 and Appendix A). 1B2CHUA shares the substitution pattern of 3BUA but contains two bulky cyclohexyl groups. The band analysis shows that in neat 1B2CHUA, the urea-group fingerprint is practically identical to that of 3BUA: the carbonyl stretching band is located at 1615 cm^−1^ (1621 cm^−1^ in 3BUA), amide II at 1526 cm^−1^ (1530 cm^−1^ in 3BUA), and the NH stretching band is centered at 3360 cm^−1^ (3345 cm^−1^ in 3BUA) (Figure 7, Table 1). The most significant difference is in the width of the NH stretching band, which is significantly broader for 3BUA. This can be explained by the state of the material—3BUA is a viscous liquid, while 1B1CHUA and 1B2CHUA are crystalline solids.

1B1CHUA is a *N*,*N*′-disubstituted urea and behaves correspondingly (Figure 7, Table 1). In the neat material, we observe a single carbonyl stretching band at 1621 cm^−1^ connected with the “ordered” urea structure. This observation is expected as 1B1CHUA is a crystalline solid. Both its amide II and N–H stretching bands split into two maxima (in-phase and out-of-phase vibration in the two NH groups) at 1530 and 1565 cm^−1^, and at 3335 and 3315 cm^−1^, respectively.

After dilution in EA (Figure 7), the urea NH groups are H-bonded to ester carbonyls, while some of the urea carbonyl bonds become free. In 1B1CHUA, this leads to the appearance of all three known carbonyl stretching bands: part of the urea remains ordered (1625 cm^−1^), and urea free from H-bonds on carbonyl (1685 cm^−1^), as well as disordered urea (~1665 cm^−1^) are identified.

In contrast, the trisubstituted ureas (3BUA and 1B2CHUA) diluted in EA display only two maxima in the carbonyl stretching region (Figure 7b): H-bonded urea (1629 and 1615 cm^−1^ for 3BUA and 1B2CHUA, respectively) and free urea (around 1651 cm^−1^). No changes are detected in the N–H stretching region, while the amide II bands shift slightly, suggesting that all NH hydrogens are H-bonded regardless of dilution.

IP has a strong and broad OH stretching band that completely obscures the NH stretching region (Figure 7a); however, the behavior of the amide II band can be observed clearly (Figure 7b). It shifts visibly for 3BUA, whereas the position stays virtually identical for the urea with bulky substituents. We can speculate that the NH groups are less sterically hindered in 3BUA, which allows for multiple H-bonding to smaller or more flexible species, such as IP. The carbonyl stretching region gives additional information (Figure 7b). After dilution, the *N*,*N*′-disubstituted urea (1B1CHUA) displays a broad band with a maximum at 1665 cm^−1^, connected with “disordered” urea. This is expected, as all carbonyl groups should be involved in hydrogen bonds, either with other urea molecules or with the solvent; therefore, we do not observe the “free urea band” around 1690 cm^−1^. On the other hand, dilution breaks the ordered structure, and consequently, the fraction of disordered urea, reflected by the increase of the broad band with maxima at 1665 and 1640 cm^−1^(Figure 7b).

The trisubstituted ureas (3BUA and 1B2CHUA) diluted in IP display a band at 1626 or 1618 cm^−1^ for 3BUA or 1B2CHUA, respectively, and a shoulder around 1600 cm^−1^. This observation supports the hypothesis that the former band is related to a similar arrangement of urea molecules, as is realized in “disordered” *N*,*N*′-disubstituted ureas (H-bonding exists but without carbonyl dipole alignment). The band at ~1600 cm^−1^ arises from H-bonding to the alcohol OH group. We can hypothesize that since the alcohol OH group is easily accessible and mobile, the dipole moments of the carbonyl and the OH in the O–H…O=C H-bonded structure can align, similarly to the carbonyl group alignment connected with “ordered” urea [27]. This would result in a shift of the carbonyl stretching band to lower wavenumbers.

## 3. Experimental Details

**Materials.** Butyl isocyanate (98%, Sigma Aldrich, Burlington, USA), *N*-butyl-*N*′,*N*′-dicyclohexyl urea (1B2CHUA, Merck, Darmstadt, Germany), *N*-butyl-*N*′-cyclohexyl urea (1B1CHUA, Merck, Darmstadt, Germany), dibutylamine (purris. Fluka, Buchs, Switzerland), toluene (dried over molecular sieve 4 Å), ethyl acetate, and isopropanol (Sigma Aldrich, Burlington, USA, spectral grade).

**Synthesis.** 1,1,3-tributylurea (synonym. *N*,*N*,*N*′-tributylurea, abb. 3BUA, Figure 3) was prepared by the following procedure: 1.033 g (10.4 mmol) of butyl isocyanate was introduced dropwise at 0 °C under argon atmosphere into a solution of 1.316 g (10.2 mmol) of dibutylamine in 2 mL of dry toluene. The reaction mixture was allowed to warm up to room temperature and stirred for 1 h. The toluene was evaporated.

The obtained dry 3BUA was used for IR study without any further purification (3BUA-dry) and diluted with ethyl acetate (solutions 3BUA—EA) and isopropanol (solutions 3BUA—IP) at various weight ratios (1:1, 1:2, 1:3, 1:4, 1:6, 1:10, and 1:15). Two additional model molecules, *N*-butyl-*N*′*N*′-dicyclohexyl urea (1B2CHUA, Figure 2) and *N*-butyl-*N*′-cyclohexyl urea (1B1CHUA, Figure 2), were analyzed in a pristine state and dissolved in IP and EA (weight ratio 1:3).

**Characterization.** The gas chromatography-mass spectrometry (GC-MS) analyses were performed on Perkin Elmer (Waltham, USA) Clarus 680 Gas Chromatograph, directly coupled with a Perkin Elmer (Waltham, USA) Clarus SQ 8 T Mass spectrometer detector. A capillary column DB-35MS (30 m, 0.25 mm, 0.25 µm) was used. The temperature program in the gas chromatograph was as follows: the initial temperature 50 °C was held for 5 min, then increased to 250 °C at 10 °C min^−1^, and then held isothermally to complete the analysis. The temperature of the injector was 200 °C. The carrier gas was helium, with the flow rate of 1 mL min^−1^. For GC-MS detection, an electron ionization system was used with an ionization energy of 70 eV, ion source temperature of 200 °C, scan mass range of *m*/*z* 15–620, and interface line temperature of 200 °C. The identification of the compounds was performed by comparing the measured mass spectra with MS spectra in NIST (version 14) and Wiley Libraries (version 9).

The Fourier transform IR spectra of all samples, both neat and in solutions, in region 4000–650 cm^−1^ were recorded using a Thermo Nicolet (Madison, WI, USA) NEXUS 870 IR Spectrometer (MCT detector; 256 scans; resolution 2 cm^−1^) using a Golden Gate ATR accessory. The solutions were covered with a Teflon cap to minimize solvent evaporation. The spectra were corrected for the carbon dioxide and humidity in the optical path, ATR-corrected, and normalized. The relevant spectral regions were deconvoluted to pseudo-Voigt bands, using an Octave script.

## 4. Conclusions

This study provides clear insight into the IR attribution of trialkyl-substituted urea groups, compared to the well-studied *N*,*N*′-disubstituted urea group, and emphasizes the significance of various H-bonding states in these systems in organic environments. It is clear that, depending on the degree of substitution, different urea types must be considered separately, as their spectral fingerprints exhibit distinct features. In our model molecule, in the tributyl urea (3BUA), the band around 1620 cm^−1^ was attributed to its H-bonded state (N–H…C=O) without the alignment of carbonyl groups, which is equivalent to the “disordered” H-bonding state in *N*,*N*′-disubstituted ureas (Figure 8). Free urea displays its carbonyl stretching around 1660 cm^−1^ in neat 3BUA and N-butyl-N′N′-dicyclohexyl urea (1B2CHUA). An equivalent to the “ordered” urea band, where the carbonyls of individual urea groups align, was not observed for 3BUA and 1B2CHUA. However, a similar band shift was likely achieved by H-bonding its carbonyl to an alcoholic OH group: the band observed around 1600 cm^−1^ in isopropanol (acting as both an H-bond donor and acceptor) was attributed to the stretching of the urea carbonyl H-bonded to isopropanol O–H.

A comparison of two trisubstituted urea model molecules, 3BUA and 1B2CHUA, showed that the nature of the substituents does not significantly affect their carbonyl stretching (Table 1). In the neat state, both trisubstituted ureas exhibited their carbonyl stretching band at nearly identical positions, (around 1620 cm^−1^), despite one being a liquid and the other a crystalline solid. This observation supports our attribution of this band to the “disordered” state and suggests that trisubstituted ureas are unable to form the “ordered” H-bonded structure. A clear distinction is observed in the amide II band, where steric hindrance from the substituents strongly influences the extent of NH-group H-bonding.

Dilution in ethyl acetate and isopropanol confirms the different origin of the band around 1620 cm^−1^ in the trisubstituted and *N*,*N*′-disubstituted urea—it is due to the “ordered” and the “disordered” state for the disubstituted and the trisubstituted ureas, respectively.

We believe that this work will facilitate more precise spectral assessment of the various states of urea-based species, contributing to a deeper understanding of materials structure and their intermolecular interactions.

## Figures and Tables

**Figure 1 molecules-30-01410-f001:**
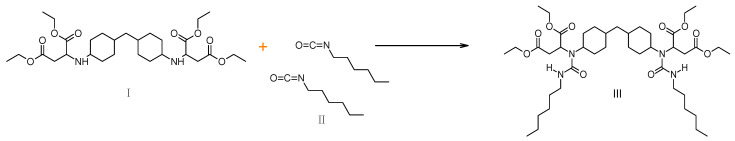
Urea groups in a model bis-aspartate urea system (III) that is formed by a reaction of the symmetrical *N*,*N*′-bis-aspartate diethyl ester (I) (systematically bis[N-(3,8-dioxa-4,7-dioxodec-6-yl)-1-aminocyclohex-4-yl]methane) with hexyl isocyanate (II).

**Figure 2 molecules-30-01410-f002:**
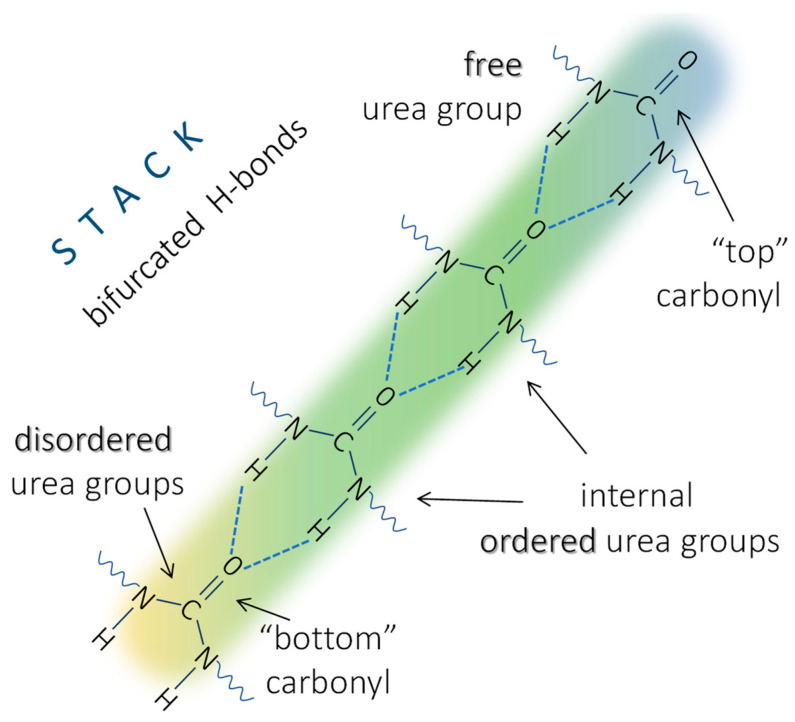
Scheme of a stack of urea molecules linked via bifurcated H-bonds, based on the work of Marcos-Fernández et al. [14].

**Figure 3 molecules-30-01410-f003:**
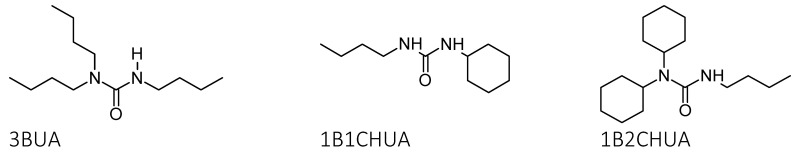
Structures of 1,1-dibutyl-3-butyl urea (3BUA), 1-butyl-3-cyclohexyl urea, and 3-butyl-1,1-dicyclohexyl urea.

**Figure 4 molecules-30-01410-f004:**
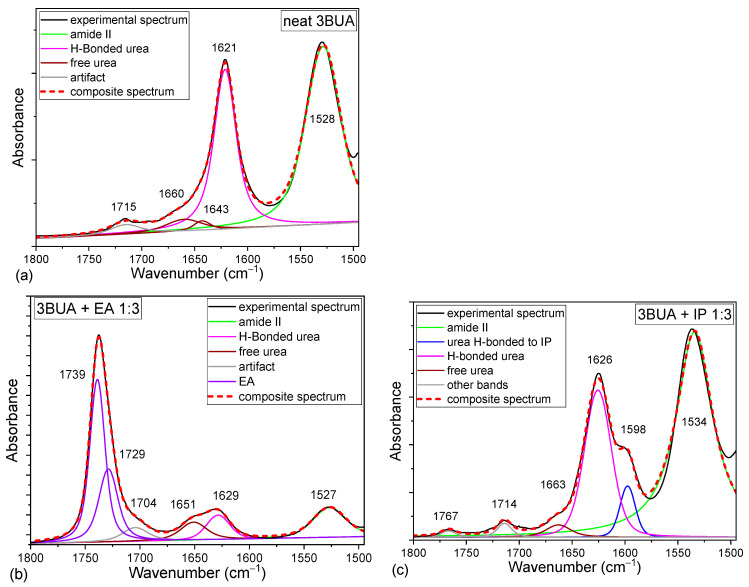
Deconvolution of the carbonyl stretching region of neat 3BUA (**a**), 3BUA + EA 1:3 (**b**), and 3BUA + IP 1:3 (**c**) to Voigt bands.

**Figure 5 molecules-30-01410-f005:**
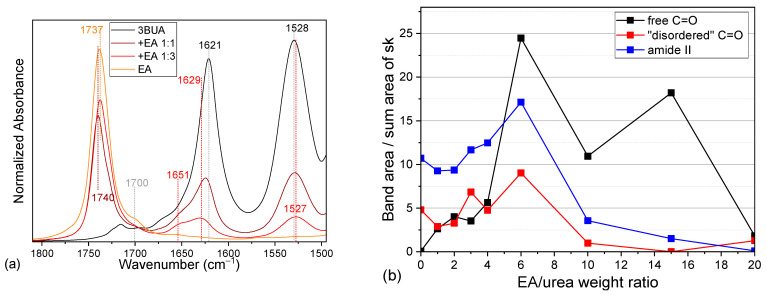
The carbonyl and amide II region of the IR spectra of neat 3BUA, neat solvents, and solutions (**a**,**c**), and the dilution dependencies of the carbonyl stretching, and amide II band areas normalized to summary area of urea skeletal bands in the region 700–800 cm^−1^ (see Appendix A); I_NH_/(I_767_ + I_742_ + I_730_) (**b**,**d**).

**Figure 6 molecules-30-01410-f006:**
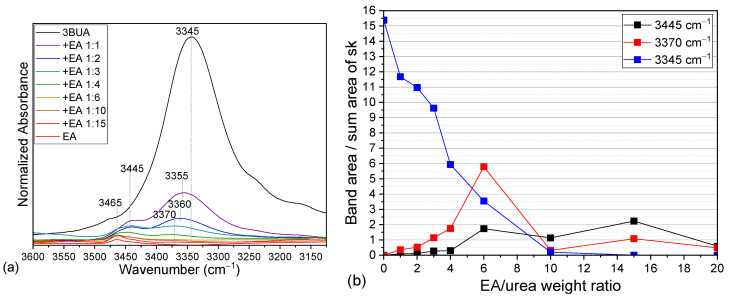
The NH stretching region of the IR spectra of neat 3BUA, neat IP, and 3BUA solutions in IP (**a**) and the dilution dependence of the NH stretching band areas, normalized to a summary area of urea skeletal bands in the region 700–800 cm^−1^ (see Appendix A); I_NH_/(I_767_ + I_742_ + I_730_) (**b**).

**Figure 7 molecules-30-01410-f007:**
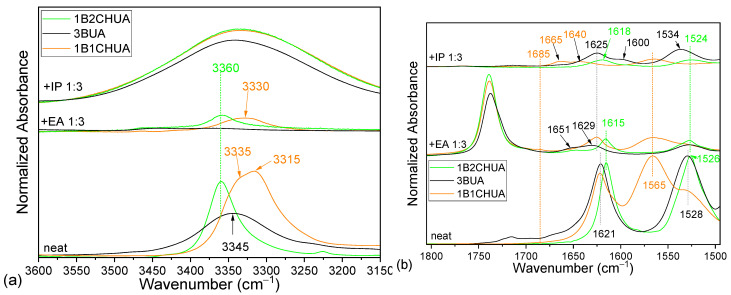
IR spectra of the three model urea molecules: 3BUA, 1B1CHUA, and 1B2CHUA, neat and diluted in EA and IP at a 1:3 weight ratio in the NH stretching region (**a**) and carbonyl stretching and amide II region (**b**).

**Figure 8 molecules-30-01410-f008:**
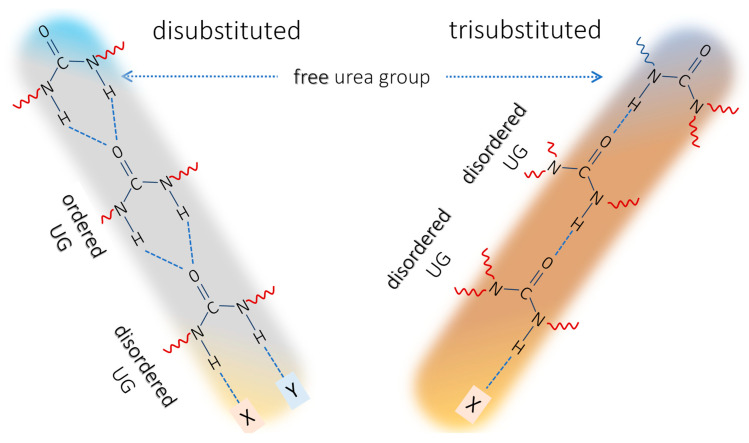
Ordering between urea groups in disubstituted and trisubstituted urea systems.

**Table 1 molecules-30-01410-t001:** The IR band positions and the attribution of *N*,*N*′-substituted and trisubstituted ureas. In addition to our model molecules, 3BUA, 1B1CHUA, and 1B2CHUA neat (roman) and in solutions (italic), typical band positions in the *N*,*N*′-substituted urea class and PU-ASPE are listed as reported in the literature. Symbols and abbreviations: NO—Not Observed, ν—stretching, δ—bending.

Disubstituted Ureas		Trisubstituted Ureas			
*N*,*N*′-sub. Urea ^i^	1B1CHUA	3BUA	1B2CHUA	PU-ASPE ^ii^	State
(cm^−1^)					
ν(N–H) Amide A					
~3300	3335, 3315	3345	3360	~3300	neat
—	*3330*	*3370, 3445*	*3360*	—	*in EA solution 1:3*
—	*NO*	*NO*	*NO*	—	*in IP solution 1:3*
ν(C=O) Amide I *“free”*					
~1690	NO	1660	NO	~1640	neat
—	*1685*	*1651*	*1651*	—	*in EA solution 1:3*
—	*1685*	*1663*	*1665*	—	*in IP solution 1:3*
ν(C=O) Amide I *“disordered”*					
~1660	NO	1621	1615	~1625	neat
—	*1665*	*1629*	*1615*	—	*in EA solution 1:3*
—	*1665, 1640*	*1626*	*1618*	—	*in IP solution 1:3*
ν(C=O) Amide I *“ordered”*					
~1635	1621	NO	NO	NO	neat
—	*1625*	*NO*	*NO*	—	*in EA solution 1:3*
—	*1625*	*NO*	*NO*	—	*in IP solution 1:3*
ν(C–N) + δ(N–H) Amide II					
~1530	1565, 1528	1528	1526	~1530	neat
—	*1565*	*1527*	*1526*	—	*in EA solution 1:3*
—	*1565*	*1534*	*1524*	—	*in IP solution 1:3*

^i^ Summarized based on the literature on urea systems of the *N*,*N*′-disubstituted type [16,17,20,21,22,27,28,30]. ^ii^ The band positions are summarized based on the literature on PU-ASPE systems [10,23,24,25], however, the attribution is revised.

## Data Availability

The original data underlying the study (FTIR spectra, deconvolution data, NMR spectrum of 3BUA, and Octave scripts) are openly available in ASPE repository at https://doi.org/10.57680/asep.0617898.

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
