# Peer review of "Hydrogen Bonding of Trialkyl-Substituted Urea in Organic Environment"

_molecules, 2025, doi:10.3390/molecules30071410_

Round 1
Reviewer 1 Report
Comments and Suggestions for Authors
Reviewer’s Report for Manuscript ID: molecules-3525051
“Hydrogen bonding in trialkyl substituted urea in organic environment”
by: Zuzana Morávková, Jiří Podešva, Valeria Shabikova, Sabina Abbrent, Miroslava Dušková-Smrčková.
This is an interesting manuscript, which has the goal of establishing the presence of different sorts of internal hydrogen bonding in model molecules for urea, either “alone” or in presence of hydrogen-bond acceptor solvents (EA) or of hydrogen-bond donating solvents (IP). The model molecules (3BUA, 1B1CHUA, and 1B2CHUA) have been synthesized de novo, with simple approaches, and have been conveniently characterized.
The main technique employed to establish the importance of the three possible types of situations is IR spectroscopy, either with poorly defined solids/gels through ATR, or with solutions in EA or in IP. And here comes my first question/comment: are the solution IR measurements run in IR (thin?) cells? Or what? Please specify.
My second question is: while in Figures 4, 5 and 6 the Authors specify the ratio (in weight? In volume?) of solvent to compound(s), what is the concentration of the solutions employed in Figure 7? We only know that the Authors employed there dilute solutions. Inter alia, I would like to say that Figure 7 is the Figure I like the most, and, if I were the Author of the paper, I would have put it at the beginning of the paper, just after Table 1, as a guide.
Third point: could the Authors provide a description (just a few words) of the band-deconvolution programs they employed? Together with the limits and possible errors.
Finally, some minor notes about the presentation: (1) The title: instead of “Hydrogen bonding in trialkyl substituted urea in organic environment”, would it not be better “Hydrogen bonding of trialkyl substituted urea and of model molecules thereof in organic environment”, or simply “Hydrogen bonding of trialkyl substituted urea in organic environment”? (2) Page 5 and everywhere else, when employed: I prefer “antisymmetric” instead of “asymmetric”, as referred to N-C-N vibration. (3) Page 10. The caption of Figure 6 looks somewhat confusing to me: instead of “…the dilution dependencies of NH stretching band areas normalized to summary are of urea skeletal bands in the region…” should it not be: “…dilution dependence of NH stretching band areas with respect to the overall (summed?) area of urea skeletal bands in the region…”?
Author Response
Comment 0: This is an interesting manuscript, which has the goal of establishing the presence of different sorts of internal hydrogen bonding in model molecules for urea, either “alone” or in presence of hydrogen-bond acceptor solvents (EA) or of hydrogen-bond donating solvents (IP). The model molecules (3BUA, 1B1CHUA, and 1B2CHUA) have been synthesized de novo, with simple approaches, and have been conveniently characterized.
Response 0: We thank the reviewer for their comments and suggestions.
Comment 1: The main technique employed to establish the importance of the three possible types of situations is IR spectroscopy, either with poorly defined solids/gels through ATR, or with solutions in EA or in IP. And here comes my first question/comment: are the solution IR measurements run in IR (thin?) cells? Or what? Please specify.
Response 1: The reviewer is right that some details of the experimental procedures were unclear. Thereby the word “weight“ was added in the dilution ratio description in the experimental section: “… at various weight ratios (1:1, 1:2, 1:3, 1:4, 1:6, 1:10 and 1:15)”; and the dilution of 1B1CHUA and 1B2CHUA was specified: 1:3. All FTIR spectra including the spectra of solutions were measured on ATR. The solutions were covered with a cap to minimize solvent evaporation. These details were added in the experimental section.
Comment 2: My second question is: while in Figures 4, 5 and 6 the Authors specify the ratio (in weight? In volume?) of solvent to compound(s), what is the concentration of the solutions employed in Figure 7? We only know that the Authors employed there dilute solutions. Inter alia, I would like to say that Figure 7 is the Figure I like the most, and, if I were the Author of the paper, I would have put it at the beginning of the paper, just after Table 1, as a guide.
Response 2: The ratios are in weight (see above). The dilutions were 1:3; this information was added to Figure 7 and its caption. The authors agree the figure 7 is important. The core information of Figure 7 was, after simplification, used in the graphical abstract, so the reader will see it soon. However, given the flow of the article, the figure 7 itself has to be where it is.
Comment 3: Third point: could the Authors provide a description (just a few words) of the band-deconvolution programs they employed? Together with the limits and possible errors.
Response 3: The Octave scripts are now disclosed as open data, https://doi.org/10.57680/asep.0617898 , the fitting parameters are also listed in the supporting information as Appendix I.
Comment 4: Finally, some minor notes about the presentation: (1) The title: instead of “Hydrogen bonding in trialkyl substituted urea in organic environment”, would it not be better “Hydrogen bonding of trialkyl substituted urea and of model molecules thereof in organic environment”, or simply “Hydrogen bonding of trialkyl substituted urea in organic environment”? (2) Page 5 and everywhere else, when employed: I prefer “antisymmetric” instead of “asymmetric”, as referred to N-C-N vibration. (3) Page 10. The caption of Figure 6 looks somewhat confusing to me: instead of “…the dilution dependencies of NH stretching band areas normalized to summary are of urea skeletal bands in the region…” should it not be: “…dilution dependence of NH stretching band areas with respect to the overall (summed?) area of urea skeletal bands in the region…”?
Response 4: Thank you for pointing these errors out. We have modified the mentioned sentences, and we believe the clarity of the message was improved.
Reviewer 2 Report
Comments and Suggestions for Authors
The manuscript by Morávková et al. reports on study of the IR spectra of 1,1-dibutyl-3-butyl urea and some related substances. In my opinion, these studies should be carried out in conjunction with theoretical methods (quantum chemistry and/or molecular dynamics). These theoretical methods are not presented in this paper. Despite this, the work presents a large systematic experimental study that is enough for publication. The described results may be useful to researchers from the physical chemistry community and organic chemistry community. Below I have indicated my comments, questions and recommendations.
- The Authors wrote a big introduction, where they described many interesting features of the field of knowledge. However, there are very few references in the text. I insist that the introduction, especially the lines 35-111, should include more relevant references.
- line 57. Tg -> glass transition temperature Tg ?
- Figure 3. I recommend rotating the figure 90 or 180 degrees so that the “top” is at the top and the “bottom” is at the bottom.
- Figure S1. I propose to make the order of the lines in the legend (top to bottom) the same as the order of the spectra on the graph.
- Lines 275-277. “Inevitably, individual urea carbonyl stretching bands (free, disordered, ~??-?? cm-1) will shift due to solvent effect, however, these effects are not in focus of this study.” Please specify the range of frequency values in the spectra.
- Apparently, it is necessary to swap figures 5 and 6, since figure 6 is mentioned earlier in the text.
- Line 299. The word "are" is redundant, isn't it?
- Figure 5. The figure is very difficult to read and understand. I suggest dividing the figures into figures dedicated to dilution with (a, c)EA and (b,d) IP.
- The Figures 5b and 6b. The authors do not describe the conclusions drawn from these data. May be it should be moved to the Supplementary Materials?
- Line 354. 3330 -> 3335 ???
- Table 1. Analyzing the numerous data presented in the article, I think I found inconsistencies. For example, ν(C—N)+δ(N—H) for pure 3BUA = 1528 (Table 1), whereas in the Figure 7 ν = 1530. Therefore, please check all the values presented in the article (in text, figures, tables, supplementary materials). I would also recommend making footnotes to table 1 for all the values for 1B1CHUA, 3BUA, 1B2CHUA, in which the numbers of Figures with the corresponding bands should be indicated.
- Figure 7. The authors chose their own color of spectrum lines for each of the substances considered. Please make the indicated values, dotted lines and arrows also of the corresponding colors.
- Lines 419-422. “We believe that as the prevalence of the different urea species has a significant impact on the properties of multiple materials, this knowledge will be of great importance for the scientific community”. In my subjective opinion, this sentence is unnecessary, since it does not have new information.
- Line 158. “GC/MS” -> “gas chromatography-mass spectrometry”
- Table S8. It is not mass spectrum.
16. Line 171. Please provide a reference to the program/script used, or describe the method in more detail (possibly in the supplementary materials).
Author Response
Comment 0: The manuscript by Morávková et al. reports on study of the IR spectra of 1,1-dibutyl-3-butyl urea and some related substances. In my opinion, these studies should be carried out in conjunction with theoretical methods (quantum chemistry and/or molecular dynamics). These theoretical methods are not presented in this paper. Despite this, the work presents a large systematic experimental study that is enough for publication. The described results may be useful to researchers from the physical chemistry community and organic chemistry community. Below I have indicated my comments, questions and recommendations.
Response 0: We thank the reviewer for their comments and suggestions. We are aware that it is a common practice to include quantum chemical calculation in in-depth spectroscopic studies, but we didn’t deem it necessary in this particular work, as the experimental data give enough proof of our conclusions.
Comment 1: The Authors wrote a big introduction, where they described many interesting features of the field of knowledge. However, there are very few references in the text. I insist that the introduction, especially the lines 35-111, should include more relevant references.
Response 1: The introduction summarizes the common knowledge on the topic, several references to review papers and textbooks were added (ref. 1-5).
Comment 2: line 57. Tg-> glass transition temperature Tg ?
Response 2: The symbol Tg was properly defined.
Comment 3: Figure 3. I recommend rotating the figure 90 or 180 degrees so that the “top” is at the top and the “bottom” is at the bottom.
Response 3: Figures 3 and 8 were rotated.
Comment 4: Figure S1. I propose to make the order of the lines in the legend (top to bottom) the same as the order of the spectra on the graph.
Response 4: The figure legend was corrected.
Comment 5: Lines 275-277. “Inevitably, individual urea carbonyl stretching bands (free, disordered, ~??-?? cm-1) will shift due to solvent effect, however, these effects are not in focus of this study.” Please specify the range of frequency values in the spectra.
Response 5: The range was specified.
Comment 6: Apparently, it is necessary to swap figures 5 and 6, since figure 6 is mentioned earlier in the text.
Response 6: Figure 5 is mentioned earlier than Figure 6, its earlier occurrence is together with Figure 4 (Fig. 4b, 5) in the paragraph just below Figure 4.
Comment 7: Line 299. The word "are" is redundant, isn't it?
Response 7: Thank you for pointing to this typo, it was meant to be „area“, not „are“.
Comment 8: Figure 5. The figure is very difficult to read and understand. I suggest dividing the figures into figures dedicated to dilution with (a, c)EA and (b,d) IP.
Response 8: The reviewer is correct, Figure 5 looks better divided into separate lines for IP and EA.
Comment 9: The Figures 5b and 6b. The authors do not describe the conclusions drawn from these data. May be it should be moved to the Supplementary Materials?
Response 9: Actually, all claims on band intensity/area decreases and increases are mainly deduced from figures 5b and 6b. Explicit references are added (replacing simple references to the whole figures 5 and 6).
Comment 10: Line 354. 3330 -> 3335 ???
Response 10: The peak center location was corrected.
Comment 11: Table 1. Analyzing the numerous data presented in the article, I think I found inconsistencies. For example, ν(C—N)+δ(N—H) for pure 3BUA = 1528 (Table 1), whereas in the Figure 7 ν = 1530. Therefore, please check all the values presented in the article (in text, figures, tables, supplementary materials). I would also recommend making footnotes to table 1 for all the values for 1B1CHUA, 3BUA, 1B2CHUA, in which the numbers of Figures with the corresponding bands should be indicated.
Response 11: The band centers were checked and corrected throughout the manuscript.
Comment 12: Figure 7. The authors chose their own color of spectrum lines for each of the substances considered. Please make the indicated values, dotted lines and arrows also of the corresponding colors.
Response 12: The figure was colored.
Comment 13: Lines 419-422. “We believe that as the prevalence of the different urea species has a significant impact on the properties of multiple materials, this knowledge will be of great importance for the scientific community”. In my subjective opinion, this sentence is unnecessary, since it does not have new information.
Response 13: The sentence in question was modified.
Comment 14: Line 158. “GC/MS” -> “gas chromatography-mass spectrometry”
Response 14: The abbreviation was properly defined in the experimental section.
Comment 15: Table S8. It is not mass spectrum.
Response 15: The captions were corrected, the actual mass spectrum was added as Fig. S9.
Comment 16: Line 171. Please provide a reference to the program/script used, or describe the method in more detail (possibly in the supplementary materials).
Response 16: The Octave scripts are now disclosed as open data, https://doi.org/10.57680/asep.0617898, the fitting parameters are also listed in supporting information as Appendix I.
Round 2
Reviewer 1 Report
Comments and Suggestions for Authors
The Authors have answered satisfactorily to this reviewer and, I guess, to other Reviewers.
Reviewer 2 Report
Comments and Suggestions for Authors
The authors made appropriate corrections.
I would recommend changing the colors of the spectra (IP 1:1 and 1:3) in Figure 5c.
I would recommend indicating the amount of EA and lambda for spectra (IP 1:3) in the figure 7a.